

# PyRINEX: a new multi-purpose Python package for GNSS RINEX data

Jinzhen Han[1], Seung Jun Lee[1], Hong Sik Yun[1], Kwang Bae Kim[1] and Sang Won Bae[2]

[1] Department of Civil, Architectural & Environment Engineering, Sungkyunkwan University, Suwon, Korea
[2] Korea Ministry of the Interior and Safety, Sejong, Korea

## ABSTRACT

Since the first receiver independent exchange format (RINEX) version was released in 1989, it has gone through several versions, making the existing software, such as TEQC, incompatible with certain later versions. This study proposes a new Python package named PyRINEX, which is developed to batch process the most generally used versions of RINEX files, namely 2.0 and 3.0. The proposed package can be used to manage and edit numerous RINEX files as well as perform a data quality check function. PyRINEX can be easily imported into any Python IDE similar to any other open-source Python package, it also makes secondary development easy for users.

# INTRODUCTION

RINEX is a standard format that designed for management and disposal of the measures generated by GNSS receiver, whatever the manufacturers of the receivers (*Gurtner & Estey, 2007*). The RINEX format is widely used in GNSS-related research and engineering projects for positioning purposes. However, not all RINEX files can be used successfully due to various reasons. Firstly, some RINEX files do not meet the format requirements due to user errors, examples include incorrectly naming the RINEX file, entering non-standard formatting in it, and misspelling receiver and antenna type, which makes them incompatible with programs such as GAMIT/GLOBK (*Herring, King & McClusky, 2010*), Bernese (*Dach & Walser, 2015*). Secondly, poor observation conditions may result in large errors in the data recorded in the RINEX file, such as clock-related errors, multipath errors, and system errors (*Karaim et al., 2018*). Therefore, it is necessary to perform data cleaning and quality checks on the dataset before using RINEX files in both research and practical engineering projects.

Existing programs for pre-processing RINEX files, such as TEQC (*Estey & Meertens, 1999*), do not work well for performing both data cleaning and quality check simultaneously. TEQC's editing functions are difficult to use for data cleaning, and its quality check function is limited by unsatisfactory visualization and inability to handle RINEX files of version 3.0 (*Gurtner & Estey, 2007*). While additional tools have been developed to complement TEQC, they mostly focus on improving one aspect of TEQC and lack comprehensive

Corresponding author
Seung Jun Lee, issue7942@naver.com

solutions for both data cleaning and quality check. Examples of such tools include a MATLAB program to process TEQC output and obtain more detailed reports of multipath effects (*Ogaja & Hedfors, 2007*), and a C/C++ program that provides increased flexibility for viewing and printing TEQC plot files (*Hilla, 2002*). Some newer programs, such as GPSQC (*Lee et al., 2012*) and G-Nut/Anubis (*Vaclavovic & Dousa, 2016*), focus on quality control and multi-GNSS data monitoring, but they also lack data cleaning and RINEX file editing capabilities.

This article introduces PyRINEX, a new multi-purpose Python package for managing and processing GNSS RINEX format. It allows for easier and quicker management of large numbers of RINEX files, as well as data cleaning and data quality checks for GPS, GLONASS, Galileo, SBAS data.

Unlike other preprocessors for RINEX files, PyRINEX first reads the native RINEX file and then stores it in JSON format into a new, easier to read and call format. This lightweight format is named LITE RINEX, and this translation will help future researchers to conduct other research on RINEX data, this has been difficult to achieve in previous programs, which have tended to provide the user with the final results without taking into account the user's requirement to actually perform more detailed calculations on the observations in the RINEX data.

Previously, programs such as TEQC have ignored the needs of researchers when dealing with large amounts of RINEX data at once, which is urgently needed in studies or specific projects dealing with tens of thousands of RINEX. PyRINEX fills this gap by supporting the user to optimize the file storage structure for a RINEX dataset. It also provides the function of self-correction of formatting errors in the header part of RINEX observation files. Compared with TEQC, the biggest difference of this part of the function is that the user does not need to specify each kind of error modification method one by one and errata each RINEX data one by one, but the program has the internal judgment to replace the manual operation, which greatly improves the efficiency. After completing the data cleaning part mentioned above, it also outputs a CSV file with the latitude and longitude coordinates of each RINEX data where it is observed, which can be imported by the user into the GIS software to complete the geo-visualization of the RINEX dataset.

In the quality check section, PyRINEX provides three outputs for the user to choose from. The first one is the quality check results of each satellite under each epoch stored as an array in Numpy, which can be helpful for some researchers who want to program further. The second is to output the array to a CSV file for researchers who only need the results. The third is the visualization of the quality check results out of the graphs, which was not available in previous programs such as TEQC.

As an open-source Python package, all of PyRINEX's code is publicly available, making it ideal for building on for secondary development to be applied to other studies by researchers in greater detail and depth.

## Structure and operation mechanism of PyRINEX

PyRINEX provides three main functions, the first is the function to read and translate into JSON format, the second is the function to manage a large number of RINEX files, and

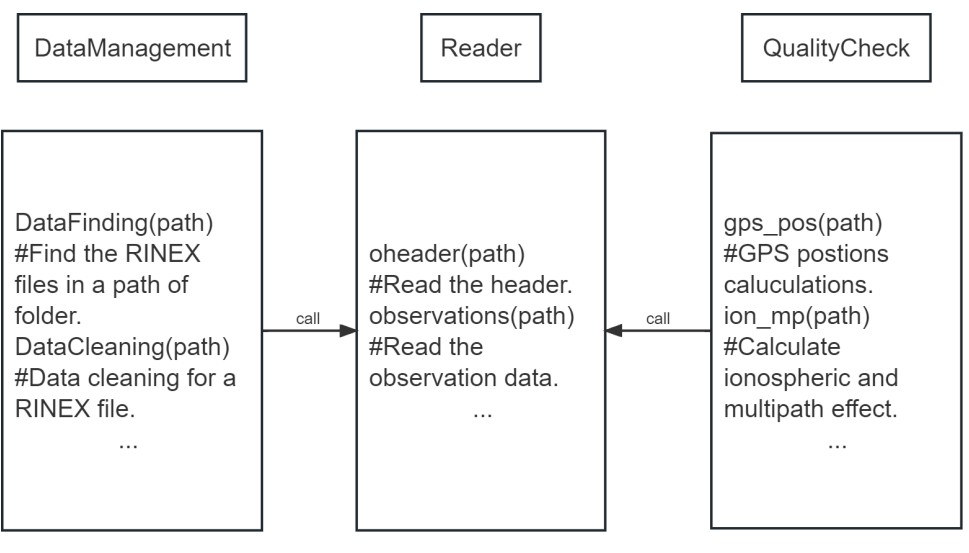

**Figure 1** Modules of PyRINEX.

the third is the function to check the data quality of RINEX files. These three functions are stored in three different modules called Reader, Data managements and Quality check.

As shown in Fig. 1, under each of the three modules there are functions written to achieve different purposes, and the common feature of all these functions is they all need absolute path of the input RINEX file as an argument. The Reader module provides the function to read RINEX files, and the other two modules need to call functions from the Reader module to implement the functions in them.

The biggest advantage of PyRINEX as a package is that the functions can be called freely by the user to control the process of the any purpose, just like any other package. Figure 2 shows how the three main functions in PyRINEX are implemented. The user can enter the absolute path of a single RINEX file to be processed into a function under the Reader module.

PyRINEX also provides the ability to allow users to batch process large amounts of RINEX data. As shown in Fig. 3, the DataFinding function implements the function of retrieving and filtering the RINEX data under a certain path. The function is implemented based on Python's os library, which is used to interact with the operating system, including file and directory operations. To use this function the user needs to enter the specified root directory, a list of keywords to filter its subfolders, and a extension representing the type of RINEX data file. After that, it will traverse all the files under the path, and then determine whether it meets the conditions, it is worth noting that even if the input extension is "08o", some files with "08O" as extension will still be output in the result list because the RINEX standard format does not specify the extension case.

The list can be used as an input parameter to other functions for subsequent processing of these RINEX data, like reading, data cleaning and quality check. This method of processing

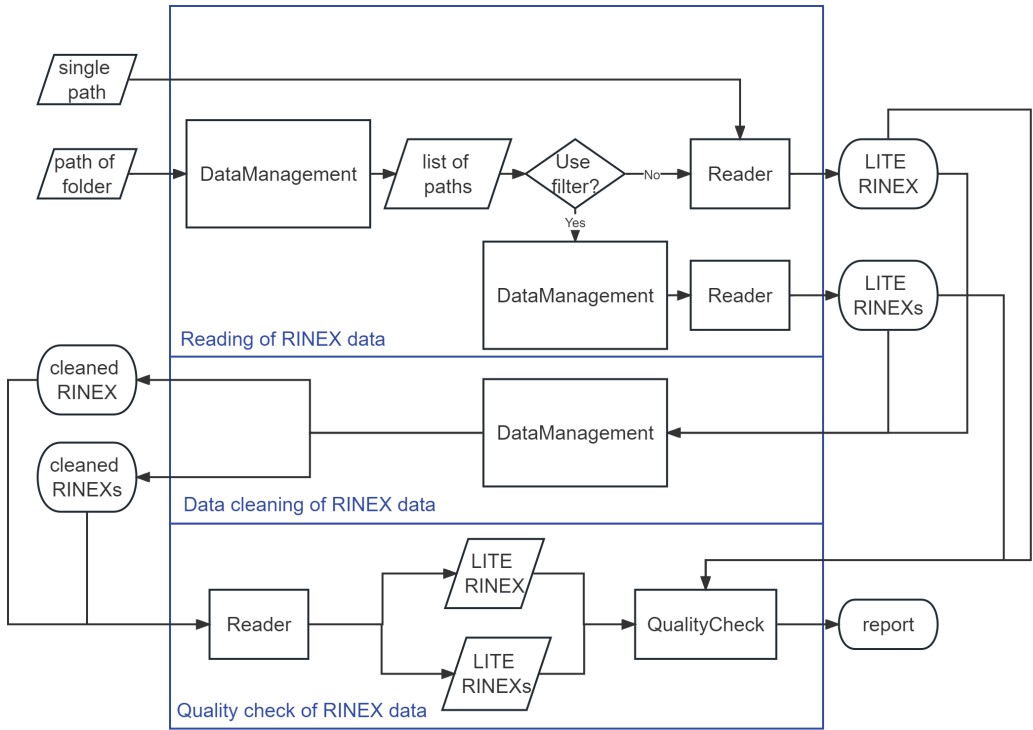

**Figure 2  Operation mechanism of PyRINEX.**

by filtering the specified RINEX files allows the user to perform more detailed operations in processing the data than is possible with other existing programs such as TEQC.

## Reading and translation of RINEX data

With PyRINEX, the RINEX data can be translated into a format called LITE RINEX, which is stored in JSON format. The native RINEX data format is difficult to read and recall due to the large differences between versions of the RINEX format and the subtle differences between even the same version of the RINEX format. Therefore, PyRINEX provides the conversion of the native RINEX data format into a new RINEX data format called LITE RINEX, which is stored in JSON format, for the sake of implementing the functions of other modules and for reading RINEX data in later studies.

As shown in Fig. 4, the header section of the RINEX observation file is transformed into more readable dictionary data. This contains some of the most important information in the header section in both categories, such as the version of the RINEX file, information about the type of observation recorded, *etc*. The second type of information is the marker name, receiver type, *etc*. The most important feature of this type of information is that it can be edited according to the user's needs, and it can be seen that this type of information is stored in a list, which is because the number of rows where these information are located is not fixed, so the line number is stored in the first item of the list for the purpose of modifying the information later on in the original file.

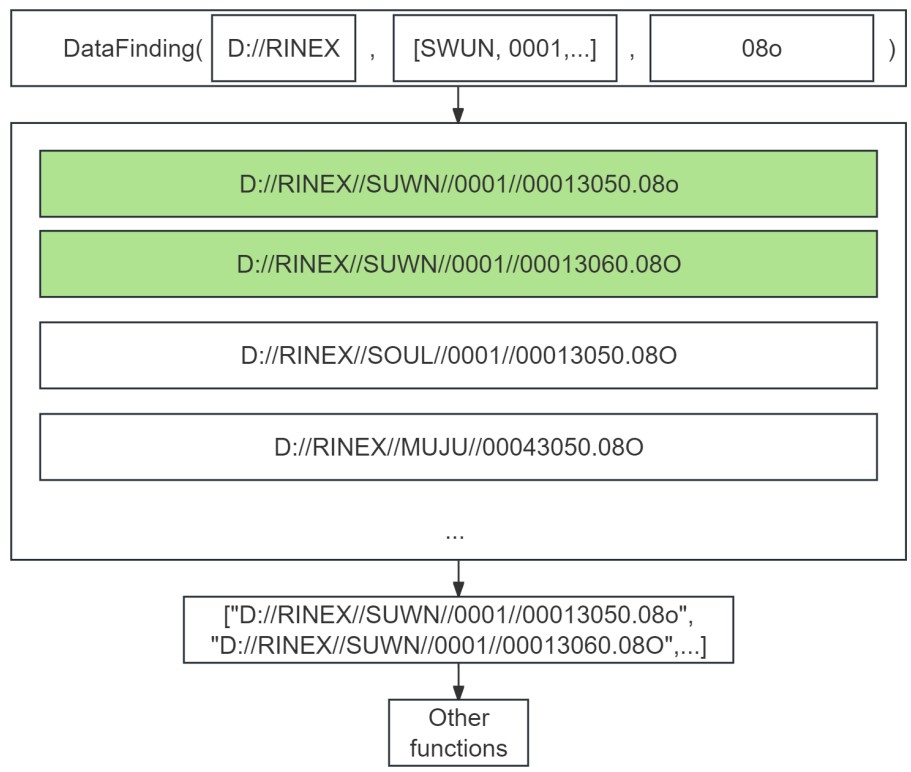

**Figure 3** Schematic diagram of the DataFinding function used to implement batch processing.

**Figure 4** The header section of the RINEX observation file after translation to LITE RINEX format.

**Figure 5** The observations section of the RINEX observation file after translation to LITE RINEX format.

**Table 1** Common errors in the header section of RINEX observation files and how to modify them.

| Error Part | Description | Correction |
|---|---|---|
| File name | 4 character station name designator does not match the marker in the file and "doy" is incorrect. | First modify the marker in the file, then recalculate the "doy", and then stitch these together into a new file name. |
| Encoding | Some non-English characters that cause the file not be stored according to the utf-8 encoding. | Remove the non-English characters and rewrite the file with utf-8 Encoding. |
| Receiver/Antenna type | Not written in a standard way | Modify it after comparing it with the content in the provided CSV file. |
| Marker name | Not following the 4 characer requierment | Modify it to 4 characters by deleting or adding. |

As shown in Fig. 5, the same logic applies to the translation of the logged portion of the observations. PyRINEX provides the ability to translate RINEX observations and GPS navigation files into LITE RINEX format. This is extremely helpful for a number of studies that aim to process GNSS observation data.

After the above steps, if the user just needs to read it, the process can be finished in this stage, but if data cleaning or quality checking is required, the LITE RINEX data processed in Reader can be fed into the corresponding functions in DataManagemt and QualityCheck for this purpose.

## Data cleaning

Only the header section of the observation file can be specified by the user; thus, only this section can cause formatting errors. However, the consequences of formatting errors in this section can be very serious, as GAMIT/GLOBK and other commonly used programs that handle RINEX files read the files strictly according to the standard RINEX file format; therefore, the slightest error can cause these programs to report errors. Table 1 presents the most frequently occurring errors and how to correct them.

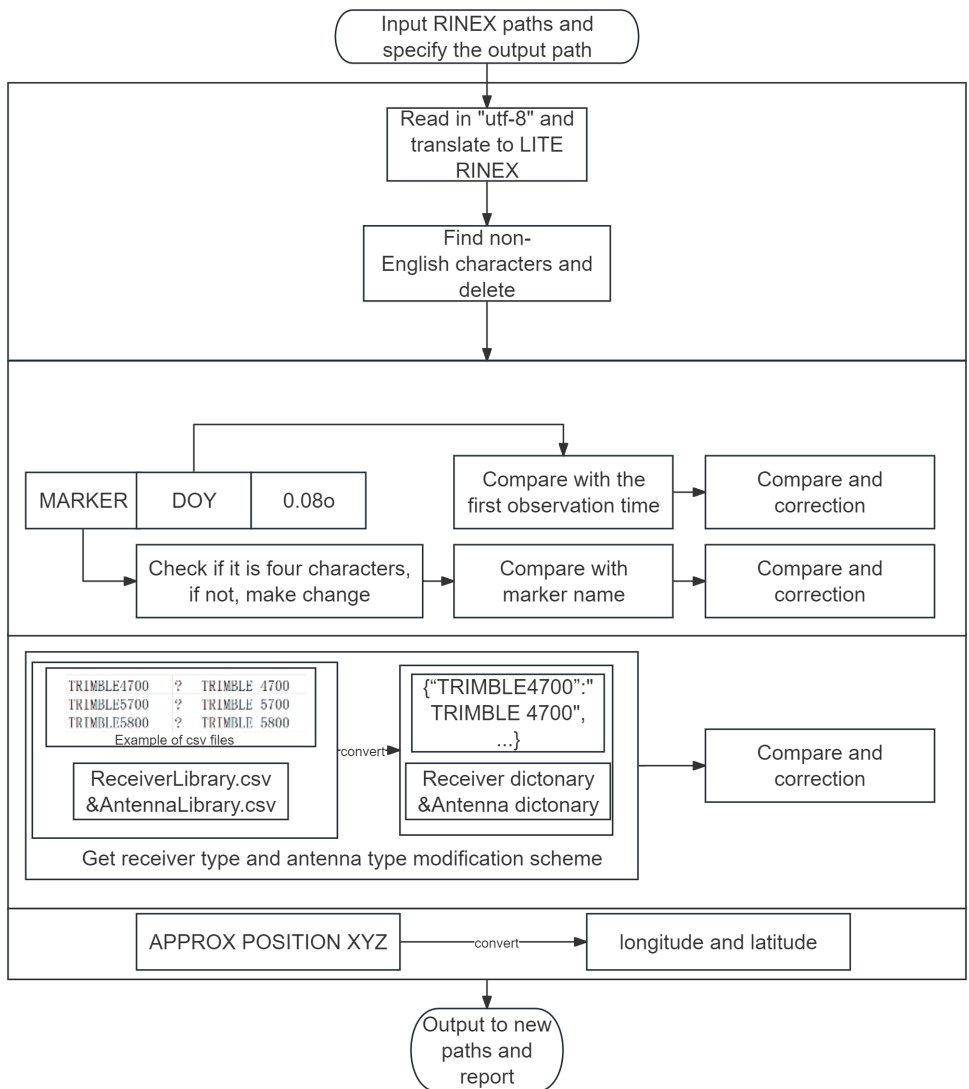

**Figure 6  Diagram of how the DataCleaning function works.**

As shown in Fig. 6, the basic logic for PyRINEX to modify the four aforementioned errors is similar. The DataCleaning function in DataManagement provide automatic *errata* for LITE RINEX after reading. It should be noted that for receiver type and antenna type corrections, the CSV files ReceiverLibrary and AntennaLibrary need to be read first, and PyRINEX will store the incorrect spelling and correct spelling as keys and values, respectively, as dictionaries in Python after reading them. After that, PyRINEX will check if there is a key in dictionray when it reads the corresponding line of the two contents, and if there is, it will replace it with the corresponding value, so that it can correct the specific contents in this way. The two CSV files can be freely edited by the user, which makes the processing of the data more customizable.

For the protection of raw data, the new RINEX file after data cleaning will be written to a specified new path, user needs to specify the root path of the output. After that, PyRINEX will use the mkdir function in the os library to create a new folder with the corresponding

**Table 2** Criteria for calculable quality checks supported in PyRINEX and their corresponding functions.

| System | Signal reception SatelliteSignalPlot (path) | Azimuth and elevation aziele (path) | Multipath effect ION_MP (path) | Ionospheric delay effect ION_MP (path) | Cycle slip effect cycleslip (path) |
|---|---|---|---|---|---|
| GPS | supported | supported | For the L1 and L2 band. | For the L1 and L2 band. | For the L1 and L2 band. |
| GLONASS | supported | unsupported | For the G1 and G2 band. | For the G1 and G2 band. | For the G1 and G2 band. |
| Galileo | supported | unsupported | For the E1 and E5a band. | For the E1 and E5a band. | For the E1 and E5a band. |
| SBAS | supported | unsupported | For the L1 and L5 band. | For the L1 and L5 band. | For the L1 and L5 band. |

"doy" name and write it to it, and then when there are RINEX data observed on the same date that are cleansed by the data, they will also be written to this folder, which can help to organize a large amount of unorganized RINEX data.

A CSV file is also output after data cleaning, which records some important information from the original RINEX file and the new RINEX file that after data cleaning (a sample report CSV file can be found in File S1). The fields in CSV file also includes longitude and latitude of where the data observed, they are converted from the approximate XYZ position from the header. The latitude and longitude recorded in the CSV file allows the file to be imported into a GIS program for visualization, which can help the user to quickly understand the geographic location of observations for a set of RINEX data.

## Quality check

The QualityCheck module in PyRINEX provides the function of quality check for the observation data and supports the processing of data from four satellite systems: GPS, GLONASS, Galileo and SBAS.

As shown in Table 2, the quality checks for signal reception, multipath effect, ionospheric delay effect and cycle slip effect are supported for all four satellite systems, and only azimuth and elevation calculations are supported only for GPS satellite systems (this is because PyRINEX does not have support for navigation data from other satellite systems). The quality check is similar to data cleaning in that the program reads the translated LITE RINEX and starts the calculation. The ability to visualize the results of all quality check calculations using the matplotlib library is provided in PyRINEX.

In the SatelliteSignalPlot function, a visualization of the satellite models received by the receiver in each time slot is provided. The function outputs a schematic diagram, which allows the user to visualize the type of satellites received during each time period and, more importantly, to know which satellites have had interruptions in the reception of their signals, which means that it is possible that poor observing conditions have triggered difficulties in the reception of the signals. Figure 7 shows schematic output of the SatelliteSignalPlot function.

Figure 8 shows schematic output of the azi_ele function. In the azi_ele function, it will calculate the coordinates of GPS satellites in the Earth-centered and Earth-fixed coordinate system of the corresponding epoch (please refer to the Appendix S1 for specific calculations). Subsequently, the approximate coordinates of the geocentric coordinate
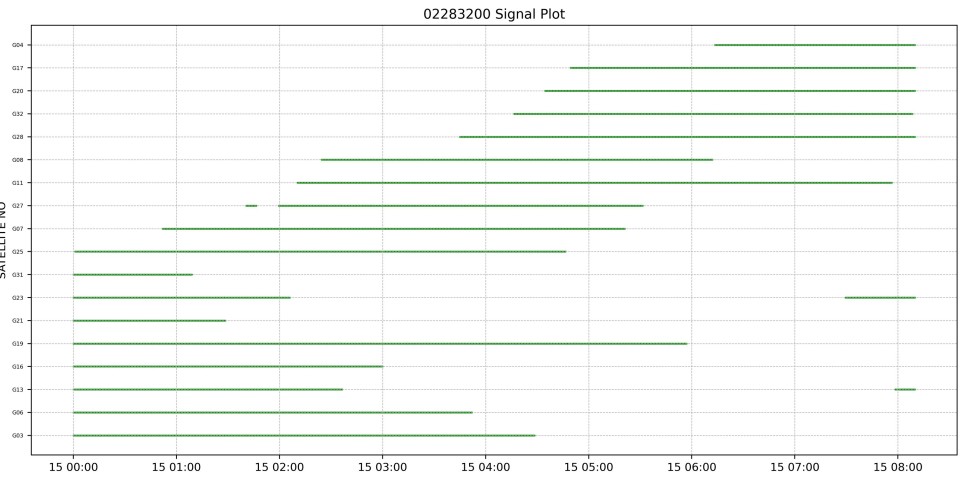

**Figure 7** Schematic output of the SatelliteSignalPlot function.

system recorded in the RINEX observation file will be converted into latitude and longitude coordinates, and the azimuth and elevation of GPS satellites in the epoch can be calculated by these two equations. Equations (1) and (2) show the calculation of azimuth and elevation, respectively. $\phi$R and $\lambda$R are the longitude and latitude of the receiver, respectively. $\Delta$X, $\Delta$Y, and $\Delta$Z are the geometrical differences between the receiver and GPS satellites.

$$\alpha = \tan^{-1}\left(\frac{-\sin \lambda_R \Delta X + \cos \lambda_R \Delta Y}{-\sin \varphi_R \cos \lambda_R \Delta X - \sin \varphi_R \sin \lambda_R \Delta Y + \cos \varphi_R \Delta Z}\right) \quad (1)$$

$$\beta = \sin^{-1}\left(\frac{\cos \varphi_R \cos \lambda_R \Delta X + \cos \varphi_R \sin \lambda_R \Delta Y + \sin \lambda_R \Delta Z}{\sqrt{\Delta X^2 + \Delta Y^2 + \Delta Z^2}}\right) \quad (2)$$

ION_MP and cycleslip functions have the same basic idea, They all start by using the Numpy library to construct a 3D array with a size of the number of recorded satellites multiplied by the number of recorded epochs multiplied by N (N being the number of results to be computed), the results of the quality checks of the satellites corresponding to each epoch are then stored in that array.

As shown in Fig. 9, assuming that a particular RINEX file has recorded signal data from four satellites observed over six epochs, such a Numpy array would be output. A zero in this array indicates that the corresponding satellite's signal was missing at that epoch, resulting in no computational results. Output like this makes it easy for users to perform more customized analysis and statistics on quality inspection results.

ION_MP function can simultaneously calculate the below two sets of values:

(a) Multi-path effect values (MP1 and MP2) that occur in L1 and L2 bands in each epoch (corresponding to the GPS system, and corresponding to other satellite systems these band names will be slightly different, but in the following they will both be referred to by L1 and L2.). The calculations for MP1 and MP2 are presented in Eqs. (3) and (4). Here, P1 and P2 denote the pseudo-range of L1 and L2, respectively, while $\Phi$1 and $\Phi$2 represent the phase measurements of L1 and L2, respectively. The parameter $\alpha$ corresponds to the square of

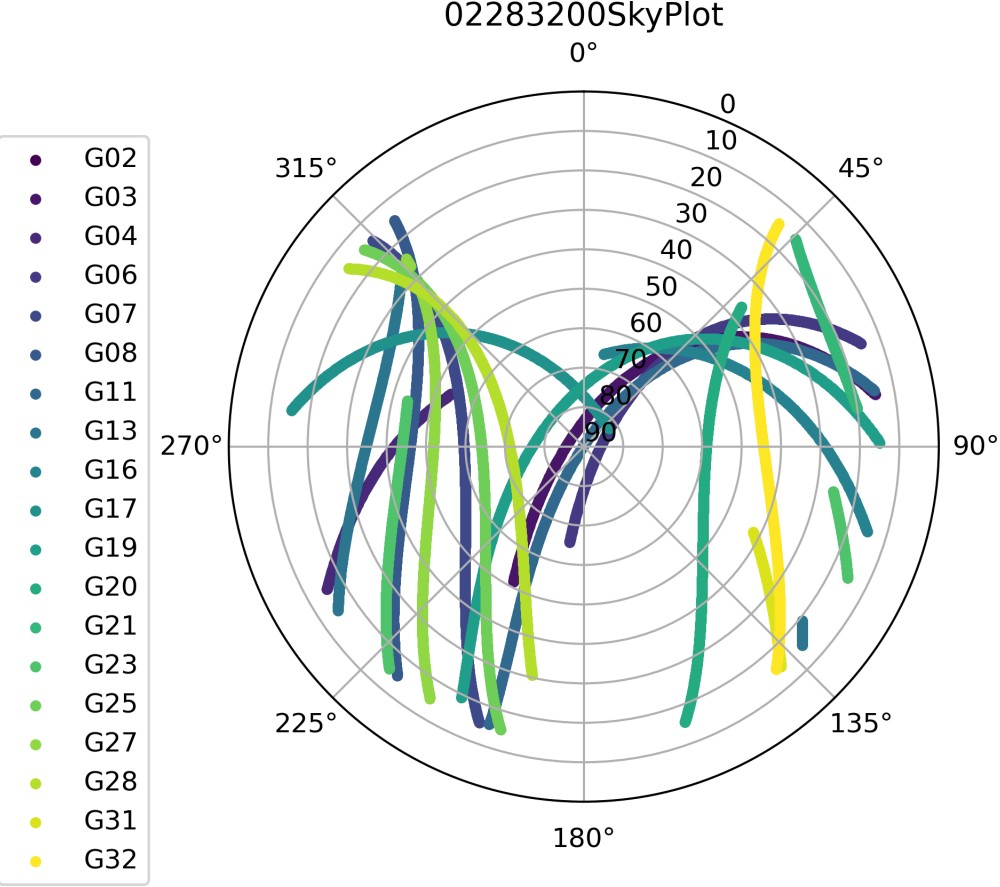

**Figure 8** Schematic output of the azi_ele function.

the ratio of frequencies between L1 and L2 (For example, for the GPS satellite system, the frequencies of the L1 and L2 bands are 1575.42 Mhz and 1227.60 Mhz, respectively, so $\alpha$ is about 1.653).

$$MP_1 = P_1 - \Phi_1 + \frac{2}{1-\alpha}(\Phi_1 - \Phi_2) \tag{3}$$

$$MP_2 = P_2 - \Phi_2 + \frac{2\alpha}{1-\alpha}(\Phi_1 - \Phi_2) \tag{4}$$

(b) Ionospheric delay (ion) and the temporal rate of change in ionospheric delay manifest during observations using a dual-frequency receiver.

The ion at one epoch is (The $\alpha$ in both Eqs. (5) and (6) are consistent with the $\alpha$ mentioned above):

$$ion = \alpha(\Phi_2 - \Phi_1). \tag{5}$$

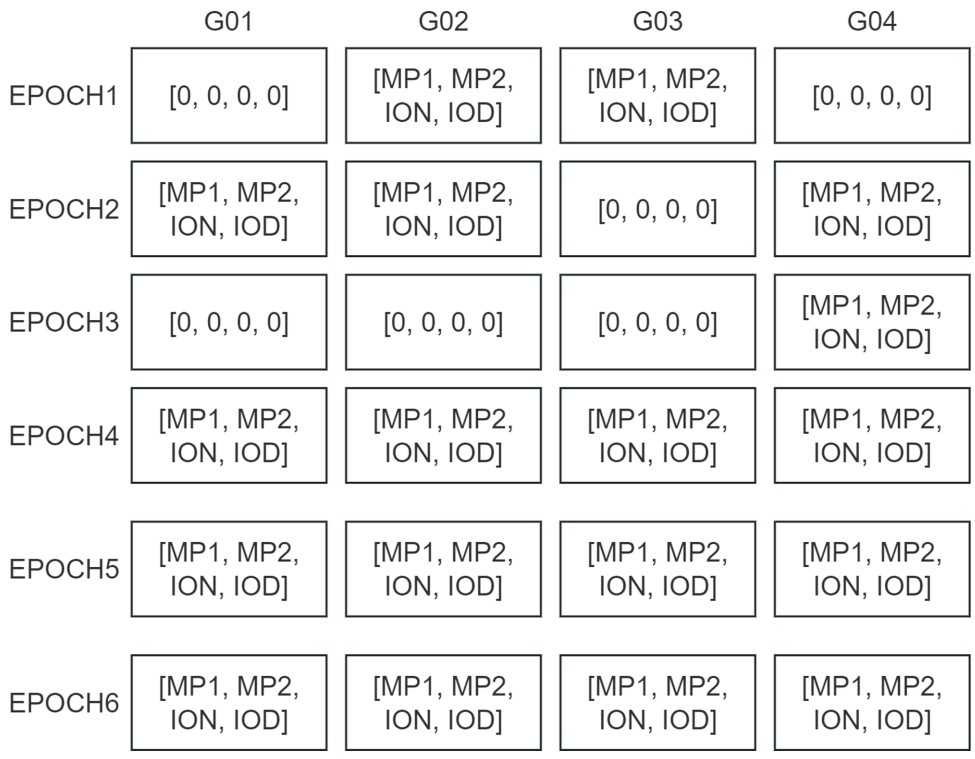

**Figure 9  Schematic of the Numpy array calculated by ION_MP and cycleslip functions.**

The temporal rate of change as the time derivative of the ionospheric delay (iod) at a specific epoch tdata is

$$iod = \frac{\alpha}{\alpha - 1} \times \frac{\left[ (L_1 - L_2)_{t_{data}} - (L_1 - L_2)_{t_{data}} \right]_1}{t_{data} - t_{data-1}}. \tag{6}$$

The cycleslip function calculates the value of the cycle slip effect, determined using the Turbo Edit algorithm established by *Blewitt (1990)*. In this algorithm, cycle slip detection relies on M-W combination (*Melbourne, 1985*) and geometry-free combination, as outlined by *Cai et al. (2013)*.

The M-W combination observation can be defined as:

$$L_{MW} = \frac{f_1 \cdot \lambda_1 \Phi_1 - f_2 \cdot \lambda_2 \Phi_2}{f_1 - f_2} - \frac{f_1 \cdot P_1 + f_2 \cdot P_2}{f_1 + f_2} = \lambda_{WL} N_{WL}. \tag{7}$$

In Eq. (7), λW L = c/(f1 -f2) ≈0.86 m (Take the GPS system as an example.) and NW L = N1-N2 are the widelane wavelength and widelane ambiguity, respectively. Where NW L can be regarded as a cycle slip test quantity, due to the fact that it is close to a constant when no cycle slip occurs.

The wide-lane ambiguity can be obtained from Eq. (7) as

$$N_{WL} = \frac{L_{MW}}{\lambda_{WL}} = \Phi_1 - \Phi_2 - \frac{f_1 \cdot P_1 + f_2 \cdot P_2}{\lambda_{WL} (f_1 + f_2)}. \tag{8}$$

The stability of the widelane ambiguity persists over time, provided that the phase observations are devoid of cycle slips. In Blewitt's TurboEdit algorithm, a recursive

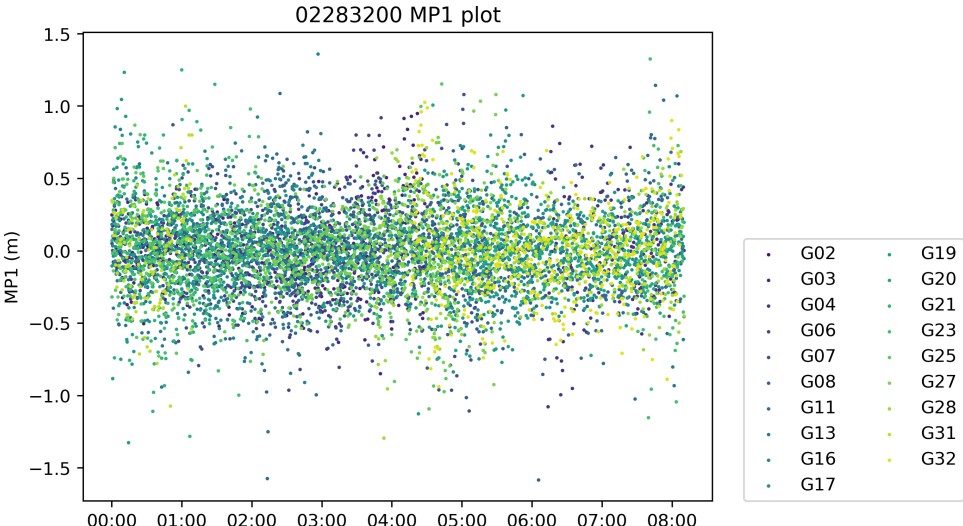

**Figure 10 Sample output from ION_MP and cycleslip functions.** A sample output of a complete quality check is available in the Supplemental Files.

averaging filter is employed for cycle slip detection utilizing the M-W combination, however, the final result provided in PyRINEX is the difference between the calculated NW L corresponding to each epoch and the NW L of the two neighboring epochs, and this difference is calculated after two difference calculations. If the difference is large it means that cycle slip has occurred, and the calculation results of each epoch can also be used to judge the quality of the observed signal of that epoch by the magnitude of the value.

Figure 10 shows a sample output MP1 computation result for the ION_MP function, where the computation result for each satellite of each epoch is plotted as a point in the result plot. In PyRINEX the ION_MP and cycleslip functions output a result plot for each of the five calculations MP1, MP2, ion, iod and cyc (with the difference in NW L between two neighboring epochs).

Unlike TEQC, PyRINEX supports quality checking of RINEX files from version 3.0 onwards. The output of quality check results are more intuitive. PyRINEX also outputs the Numpy array shown in Fig. 9 in CSV mode (A sample output CSV file is found in supplemental files along with the quality check results), which is more convenient for users who are more accustomed to working with CSV data. These CSV files are especially important when using Excel or similar software if users who want to produce more customized visual quality check results.

## Testing of PyRINEX

To test PyRINEX, in this article applied it to a research project from the Korea National Geographic Information Institute (NGII), in which NGII attempted to organize data from GPS observations taken at unified control points in Korea over the past decade or so and to apply it to a number of follow-up studies. In this article, use the data from 2008 as an

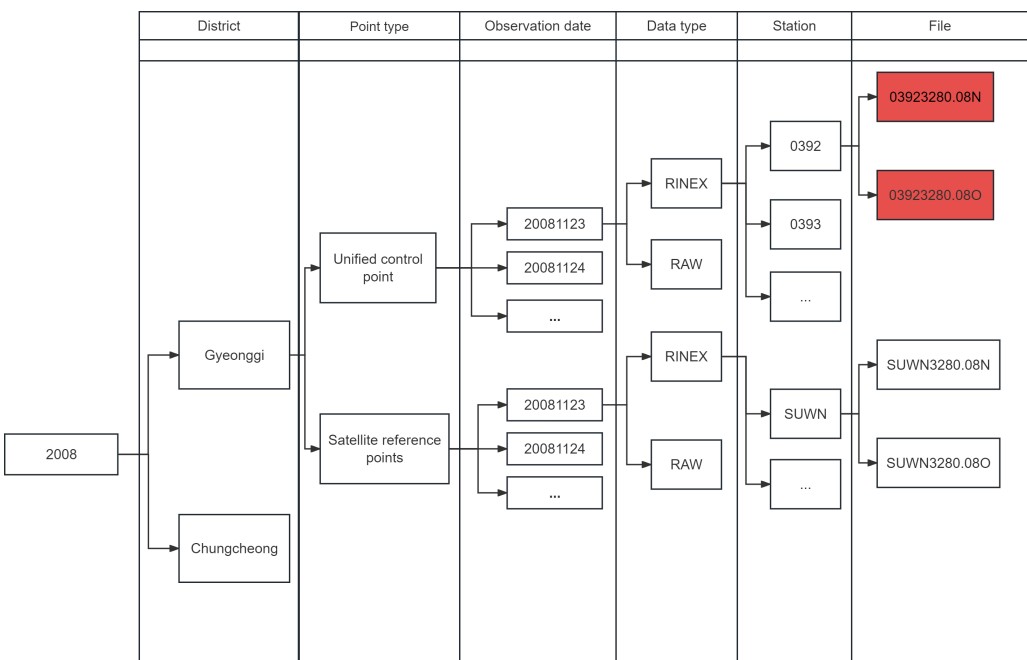

**Figure 11** Structure of the dataset.

example, which was chosen because the dataset from that year is relatively old, and at that time the dataset from that year was of lower quality due to unskilled surveyors.

As shown in Fig. 11, the biggest problem encountered in the first step of processing the data is that the structure of the data set is very complex, with six levels of paths, which are initially divided into larger administrative areas, and the biggest problem is that at the second level they are divided into two types of unified control points and satellite reference points. The former is a total of 5,588 measurement reference points spread throughout Korea, GNSS observations will be made at these points from time to time. The latter is a total of 92 satellite reference points on which GNSS observation receivers are installed for 24-hour observation (The number of data in the dataset is less than the number of sites because not every site is observed every year). If the researcher just use file manager simply searches by the suffix "08o", the results of observations at satellite reference points that are not needed will be mixed in.

We first filtered this dataset using the file filter in DataManagement module and wrote it under a new path, performing an errata on the file format at the same time in the process. As shown in Fig. 12, the new dataset has much cleaner paths and is easier to call.

As shown in Table 3, the frequency of occurrence of various types of formatting errors in the dataset can be counted after data cleaning. It can be seen that the percentage of non-English characters errors is the highest, a total of 411 files out of 898 files have wrong file names, accounting for 45%. In total, there were 663 files with one or more formatting errors, accounting for 73% of the total, and more than half of the files could not be directly applied to programs such as GAMIT, Bernese, and so on. In the past, these errors would

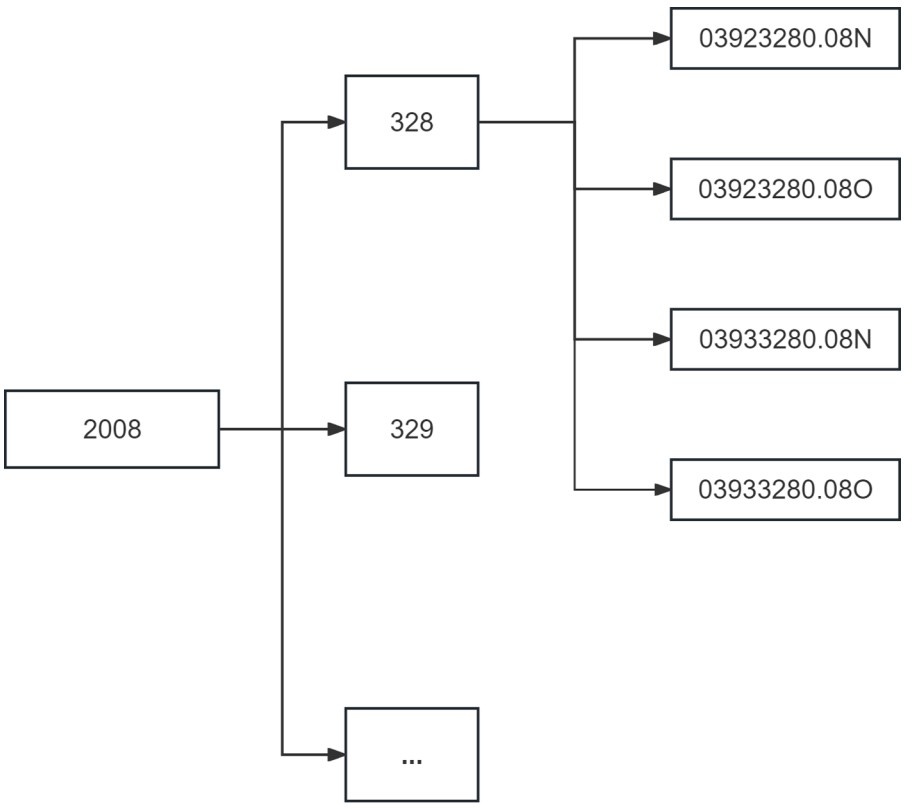

**Figure 12** Structure of the new dataset.

**Table 3** Statistics on the results of data cleaning.

| Error part | Filename | Non-English characters | Receiver type | Marker name | Total formatting errors |
|---|---|---|---|---|---|
| Amount | 285 | 411 | 227 | 2 | 663 |
| Percentage | 32% | 45% | 25% | 2% | 73% |

have been corrected manually by a researcher spending a lot of time, but it only takes a minute to process about a thousand files in PyRINEX.

As shown in Fig. 13, the spatial distribution of observations in this dataset can be analyzed after inputting the data cleaned report CSV file into the GIS software. It can be clearly seen that the distribution of the stations with problems in RINEX format has a clear distribution tendency according to administrative divisions, this is because all the observations are divided into administrative regions for different operators to carry out observations, and if a surveyor keeps the same error habit when generating RINEX files, it will lead to the problems of the RINEX files of a region.

Subsequently, in order to test the reliability of the satellite data quality check, data obtained after observations at a group of sites in Korea in November of that year were chosen as a test. Figure 14 shows the sites that were chosen for testing.

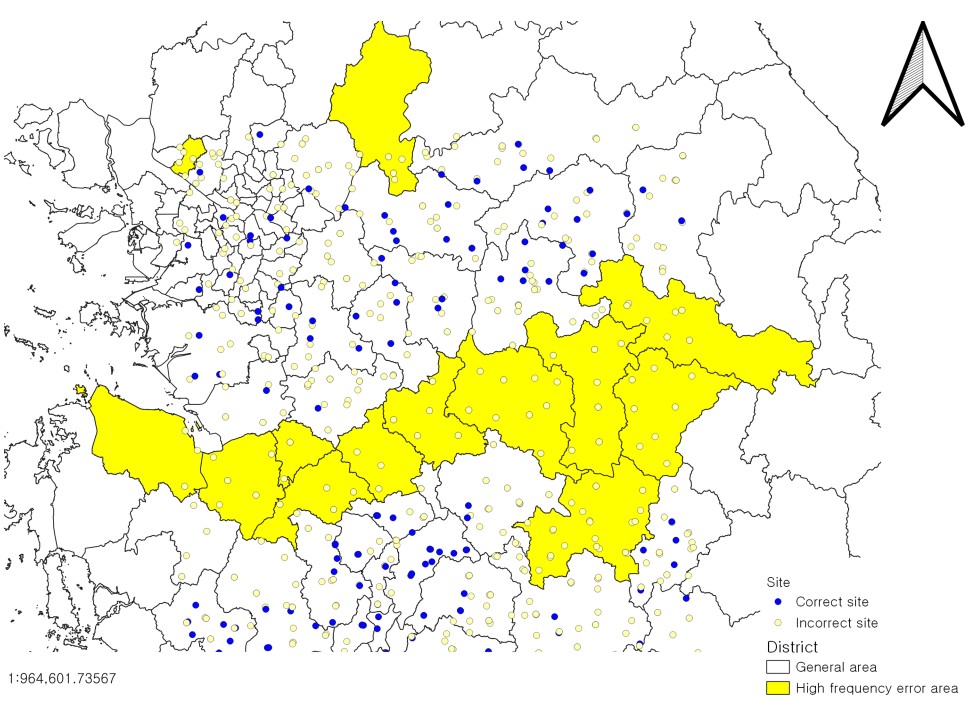

**Figure 13  Distribution map of observation sites.** Base map source: National Spatial Data Infrastructure Portal.

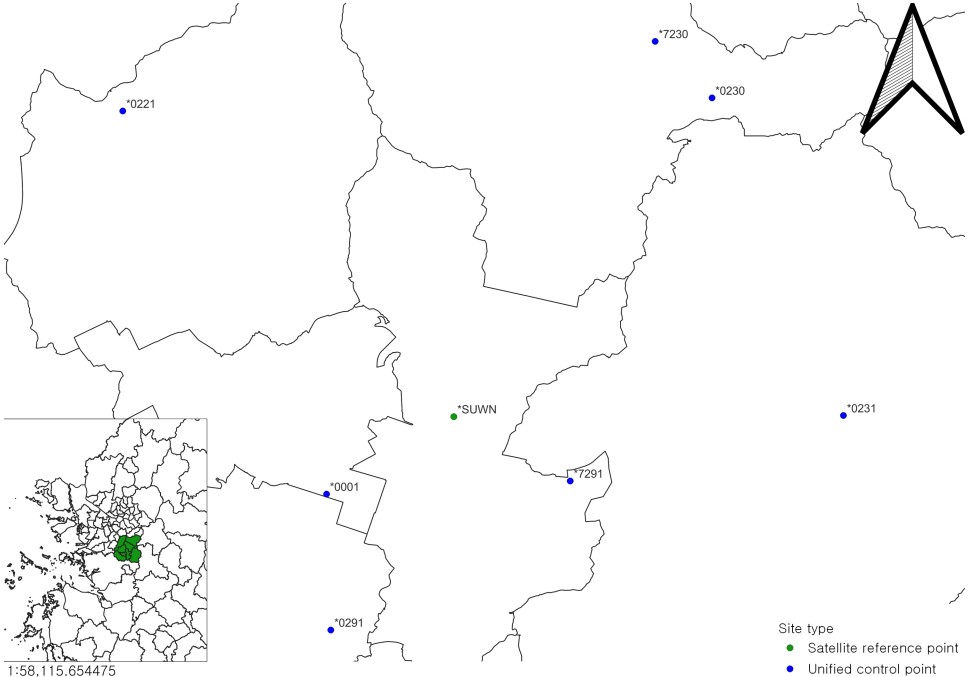

**Figure 14  Sites chosen for testing.** Base map source: National Spatial Data Infrastructure Portal.

**Table 4  Quality check results for the SUWN site.**

| PARAMETERS | MP1(m) | MP2(m) | CYCLESLIPS | ION | IOD |
|---|---|---|---|---|---|
| Criteria | <0.35 | <0.46 | <±0.59 | <±6.58 | <±0.05 |
| Allowance | 90% | 90% | 90% | 90% | 90% |

**Table 5  Results of quality check.**

| SITE MARKER | MP1 | MP2 | ion | iod | cyc |
|---|---|---|---|---|---|
| 0001 | 91.4% | 90.2% | 99.9% | 93.5% | 77.9% |
| 0221 | 90.4% | 87.2% | 97.7% | 92.3% | 60.95% |
| 0228 | 91.1% | 90.9% | 97.2% | 93.7% | 71.3% |
| 0230 | 90.8% | 91.1% | 99.9% | 93.9% | 72.4% |
| 0231 | 88.7% | 86.6% | 95.5% | 94.0% | 78.0% |
| 0291 | 90.5% | 90.3% | 97.7% | 93.8% | 70.1% |
| 7221 | 61.3% | 67.4% | 100.0% | 93.3% | 30.9% |
| 7230 | 65.2% | 66.6% | 92.9% | 92.3% | 43.0% |
| 0184 | 68.6% | 70.2% | 99.7% | 96.0% | 37.7% |

The selected sites are near Suwon, Gyeonggi-do, because there is a site's marker named "SUWN", which was set up by NGII, with a good receiver and good observation condition, and is part of the IGS global network, so the quality of the data obtained from the station can be trusted.

In this experiment the observation at the SUWN site were first quality checked and the results of the quality check were used as the signal quality standard for that time period. Because there is currently no common standard for quality check indicators. The data for the SUWN site are available in the Crustal Dynamics Data Information System (CDDIS), and since this database provides daily data, this RINEX data was first clipped to set the time period of its observations to coincide with the observations made at the unified control point.

The results of the quality checks performed on the SUWN sites are shown in Table 4. The results of all the quality checks in this study were limited to a distribution range of 90%, and then the upper and lower limits of this distribution range were obtained and designated as the criteria for the quality check results when observations were made in the area during that time period.

The results of the quality check of the RINEX data in the test set are presented in Table 5. It can be noticed that for the four results MP1, MP2, ion and iod at most of the sites, the data obtained from the observations at these unified control points are not much worse than those at the SUWN site, but the results for cyc are clearly very different, which is due to the differences in the observation conditions.

After that, the best quality check resolution 0001, 0230 and 0231 were taken as a set of data and the worst 7,221, 7,230 and 0184 were taken as a set of data. These two sets of data were subjected to network adjustment together with the data from SUWN sites, and these data needed to be designated as control points.

**Table 6  Comparison of SUWN coordinates calculated by two network adjustments and coordinates provided by NGII.**

| Source | Longitude | Latitude |
|---|---|---|
| Provided by NGII | 37°16′31.8529 N | 127°03′15.2638 E |
| Adjustment with bad data | 37°16′31.86561″N | 127°03′15.27481″E |
| Adjustment with good data | 37°16′31.86300″N | 127°03′15.25229″E |

Table 6 shows the results of the two network adjustment, the exact coordinates of SUWN in the first row is provided by NGII. The difference between the two results and the exact coordinates is then calculated using the Haversine formula as follow

$$Distance = 2 \cdot R \cdot \sin^{-1} \sqrt{\sin 2\left(\frac{\Delta Latitude}{2}\right) + \cos(Latitude_1) \cdot \cos(Latitude_2) \cdot \sin 2\left(\frac{\Delta Longitude}{2}\right)} \quad (9)$$

where $\Delta$ Latitude and $\Delta$ Longitude are the latitude difference and longitude difference between the two points, latitude1 and latitude2 are the latitudes of the two points, respectively, and R is the radius of the Earth, which is taken to be 6,371 km.

The final result obtained was that when the network adjustment was done with the worse sites, the obtained SUWN's stations differed from the actual coordinates by about 369 m, compared to about 6 m when it was done with the better sites. This proves that the quality check function of PyRINEX is reliable and can distinguish between good and bad quality RINEX data.

# CONCLUSIONS

A Python package was developed for preprocessing GNSS RINEX files, including data cleaning and quality checking for satellite data, and supporting batch processing of large amounts of data. This package has the following features:

1. It can read the most common RINEX file versions (2.0 and 3.0) and automatically extract important information to a new JSON format to make it easier for subsequent users to analyze RINEX files.
2. It can handle errors in file format caused by user mistakes in the header section of RINEX observation files, allowing normal processing by programs like GAMIT.
3. It supports quality checking of GPS, Galileo, GLONASS and SBAS satellite data, including elevation and azimuth (just for GPS sensing data), MP1, MP2, ion, iod, and cyc, with visualization of the quality check results. To support further research, quality check results can be exported to a CSV file.

The datasets obtained from the NGII observations in 2008 were processed, and it was found that the program can be used in practical research and engineering projects to efficiently process numerous RINEX files for data cleaning. Accordingly, its quality check function can be used as a tool to assess the quality of data in practical GNSS observation projects and positioning.

At this stage, there are not many free programs that can be used to perform quality checks on RINEX data, and the most prestigious one, TEQC, has ended its life cycle (EOL)

after the final version was released on February 25, 2019, and its last version is still not able to process RINEX data from version 3.0 and above. Researchers are faced with the problem of either continuing to use unmaintained TEQC or paying hundreds or thousands of dollars for commercial software. At the same time, PyRINEX offers the ability to manage large amounts of RINEX data in a way that existing programs, including TEQC, do not.

PyRINEX demonstrates its superiority when dealing with tens of thousands of RINEX data for nation-based studies. The built-in automatic conditional judgment function that enables data cleaning to be performed automatically is also an innovative point. The ability to output multiple data types with its quality checking feature also allows researchers to conduct more detailed and in-depth studies. PyRINEX as the next potential alternative, which is open source and makes all code public. This package, written 100% in Python, has a good readability that allows researchers to quickly understand it and develop it. It can be used in any IDE like Numpy or matplotlib and other prestigious libraries. This means that the package is very easy to use. Its expandability allows it to be used for a variety of different purposes in research and practical engineering.

### Funding
This research was supported by the National Research Foundation of Korea (NRF) grant funded by the Korean Government (MSIT) (no. 2021R1A2C201231913). The funders had no role in study design, data collection and analysis, decision to publish, or preparation of the manuscript.

### Grant Disclosures
The following grant information was disclosed by the authors:
National Research Foundation of Korea (NRF) by the Korean Government (MSIT): 2021R1A2C201231913.

### Competing Interests
The authors declare there are no competing interests.

### Author Contributions
- Jinzhen Han conceived and designed the experiments, performed the experiments, analyzed the data, performed the computation work, prepared figures and/or tables, authored or reviewed drafts of the article, and approved the final draft.
- Seung Jun Lee performed the experiments, prepared figures and/or tables, and approved the final draft.
- Hong Sik Yun conceived and designed the experiments, authored or reviewed drafts of the article, and approved the final draft.
- Kwang Bae Kim performed the experiments, authored or reviewed drafts of the article, and approved the final draft.
- Sang Won Bae analyzed the data, authored or reviewed drafts of the article, and approved the final draft.

## Data Availability

All source code is avaialble at GitHub and Zenodo. The data is available at Zenodo (PyRINEX_TestData):

https://github.com/geumjin99/PyRINEX

geumjin99. (2023). geumjin99/PyRINEX: v3.0.1 (v3.0.1). Zenodo. https://doi.org/10.5281/zenodo.10288240.

## Supplemental Information

Supplemental information for this article can be found online at http://dx.doi.org/10.7717/peerj-cs.1800#supplemental-information.

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
