# Peer review of "PyRINEX: a new multi-purpose Python package for GNSS RINEX data"

_PeerJ Computer Science, doi:10.7717/peerj-cs.1800_

## Round 0.1 · original submission · Major Revisions

Both Reviewers provide some interesting input to improve the paper. In particular I would like to see addressed the major points made by Reviewer 2.

That is:

- Provide more motivation and justification for PyRINEX compared to existing tools like TEQC. What specific limitations is it trying to address? (Needed)

- Include more technical details on the software implementation - architecture, key algorithms, sample code snippets, etc. (This can be addressed briefly if published under an open source license. However, the architecture may be provided - not just the "Moudles" (modules?) and see also Reviewer 1’s comment on providing information about applied algorithms for “quality check processes (signal reception, multipath effect, ionospheric delay effect and cycle slip effect)”)

- Expand testing to more varied RINEX datasets - different versions, diverse observation conditions, multiple geographic locations etc. (Needed: test your software with at least one more dataset not from Korea, and ideally a different Rinex version, and report briefly if this test has been successful - you can document the details about this elsewhere - or in a supplement)

- Justify the quality check criteria with references or through statistical analysis on expected value distributions. (Needed)

- Discuss the potential impact and significance of PyRINEX more in the conclusion - how it can benefit the GNSS community compared to current tools. (Needed)

- Perform comparisons to other existing tools like TEQC on metrics like runtime, quality check accuracy, etc. to demonstrate quantitative advantages. (Not necessary, but nice to have. Could go into an Appendix)

- Release PyRINEX as an open-source tool to allow community access, contribution and further validation. (I highly suggest this too, i.e. say where it is available and what license applies)

Please also address the issues found by Reviewer 1 regarding documentation and use with data from other sources. I find these very critical for publishing your software. Because if this works only for your data, then what is the point of publishing the results? Hence, I also would like to see a new section that explains the availability of the software and specifies the license used.

Reviewer 1 ·

Basic reporting

In my opinion the paper can be published after consideration of some major corrections:

- The biggest problem is that a basic manual for installing and using the software is not provided among the supplementary material, so it is not possible to test the software to extract conclusions. Likewise, it is necessary to provide test files with English names along with the results of their processing to verify that the installation and processing of these files by a user are the same as those expected by the authors. These two aspects are basic for any software that wants to be published.

- The DataManagement module only checks the header of the files and cleans it of possible errors, but it would be interesting if it could incorporate more basic editing operations, for example decimate an observation RINEX file (that is to increase the time among the epochs), generate a common navigation file from the individual navigation file of stations, etc. Likewise, it would be interesting to incorporate a functionality that corrects header errors and returns a RINEX file with the corrected header ready to be used by other “classic” processing programs. If these functionalities are added, editing capabilities will be added that, reading the abstract and the introduction, are expected to be incorporated into the software.

- The quality check processes (signal reception, multipath effect, ionospheric delay effect and cycle slip effect) should be explained in some detail in the manuscript. Do the elevation and azimuth calculations refer to the elevation and azimuth of the satellites? Why have they not been able to be implemented in the GALILEO, GLONASS and BEIDU constellations?

-lines 137-141. The graphic result of the quality check is only a sky plot? An example of the CSV file generated along with its explanation is necessary here (and in the software manual).

-Line 161. Here the idea that the DataManagement module contains more routines apart from the header cleaning routine is introduced, it is necessary that all the functionalities are detailed in the explanation section of this module. “filterfilter in DataManagement moudle”, do you mean “filter in DataManagement module”?

-lines 181-182. Please specify how many stations and files per station have been taken into account in this test.

-Line 192. Do you mean nine RINEX files one for every one of the nine stations?

-Line 195. 7 or 9 sites?

-Figure 9 and corresponding explanation in the text. Please use the standard deviation of the North, East and Up components instead of the semi-axes of the error ellipses, so the result is much more intuitive for any reader, even those not specialized in geodesy and GNSS.

-The last bibliographic reference, at least in the downloaded pdf file for review, is not entered correctly.

Experimental design

See previous comments on basic reporting

Validity of the findings

See previous comments on basic reporting

Cite this review as

Reviewer 2 ·

Basic reporting

This manuscript introduces PyRINEX, a new Python package for managing and processing GNSS RINEX format data. The authors highlight limitations of existing RINEX processing tools like TEQC in handling newer RINEX versions and performing comprehensive data cleaning and quality checks. PyRINEX provides three main functions - reading/converting RINEX data into a more accessible JSON-based format called LITE RINEX, bulk managing large volumes of RINEX files, and performing quality checks on RINEX observations. The package is designed to be easily imported into Python and supports secondary development. The authors test PyRINEX on a real-world RINEX dataset from the Korea National Geographic Information Institute, demonstrating its utility in efficient data cleaning of formatting errors across numerous files. They also validate the quality check capabilities by showing poorer positioning accuracy when using lower quality RINEX data as control points.

The main concerns are:

- The introduction provides background on RINEX and existing tools, but does not sufficiently motivate the need for a new tool like PyRINEX. More justification is needed on the limitations of existing tools like TEQC and why a new Python-based tool is necessary.

- The description of the PyRINEX software is quite high-level and lacks technical details. More implementation specifics on the code structure, algorithms, etc. would strengthen the paper.

- The testing methodology is limited to a single dataset from 2008. Testing on more diverse and recent RINEX data could better validate the capabilities of PyRINEX.

- The quality check criteria in Table 4 seem arbitrary. Some justification or references are needed for the proposed thresholds.

- The conclusion mainly summarizes the features of PyRINEX. More discussion is needed on the significance and potential impact of the tool.

The major comments are:

- Provide more motivation and justification for PyRINEX compared to existing tools like TEQC. What specific limitations is it trying to address?

- Include more technical details on the software implementation - architecture, key algorithms, sample code snippets, etc.

- Expand testing to more varied RINEX datasets - different versions, diverse observation conditions, multiple geographic locations etc.

- Justify the quality check criteria with references or through statistical analysis on expected value distributions.

- Discuss the potential impact and significance of PyRINEX more in the conclusion - how it can benefit the GNSS community compared to current tools.

- Perform comparisons to other existing tools like TEQC on metrics like runtime, quality check accuracy, etc. to demonstrate quantitative advantages.

- Release PyRINEX as an open-source tool to allow community access, contribution and further validation.

The minor comments are:



1. What specific limitations of existing tools like TEQC does PyRINEX aim to address? More justification is needed on why a new tool is necessary.

2. Can you provide more technical details on the software implementation? For example, code structure, key algorithms, sample snippets etc.

3. Have you tested PyRINEX on more diverse RINEX datasets beyond the 2008 NGII data? Testing on varied data would further validate its capabilities.

4. How did you determine the quality check criteria thresholds in Table 4? Are they based on statistical analysis or existing standards?

5. Can you benchmark PyRINEX against TEQC or other tools on metrics like runtime, quality check accuracy, etc. to demonstrate quantitative improvements?

6. Do you plan to release PyRINEX as an open source tool for community access and contribution? This could aid further testing and adoption.

7. Can you provide more discussion on the potential real-world impact and significance of PyRINEX for the GNSS community?

8. Is PyRINEX able to handle newer RINEX versions beyond v3.0? What about emerging formats like RTCM 3?

9. Does PyRINEX allow parallel batch processing across multiple files/cores? This could further improve performance.

10. Can the quality check visualizations be customized or expanded to user needs? More flexible output options would be useful.

Experimental design

no comment

Validity of the findings

no comment

Additional comments

no comment

Cite this review as

---

## Round 0.2 · Minor Revisions

Dear Authors,

I had a read over your response letter and the changes you did seem good - and almost satisfactory to me (only reading the letter). However, a previous reviewer revised quickly in more detail your second version, over the weekend, and I hope you can address his issue of not being able to install and run the software, as it seems. I copy his points:

-In the second version of the paper, there are no files to check the software.
-The manual lacks of the procedure to introduce the path information to run the so4ware or to run the so4ware itself (using the terminal, an IDLE...which order...how to introduce input/output files...)
-csv files with antenna and receivers are not provided (at least for the ones of the provided test).
-During the installation process the following error occurs:
error: Setup script exited with error: Failed to download any of the following: ['https://downloads.sourceforge.net/project/freetype/freetype2/2 .6.1/freetype-2.6.1.tar.gz', 'https://download.savannah.gnu.org/releases/freetype/freetype- 2.6.1.tar.gz'].

- As I said, it is necessary to provide test files along with the results of their processing to verify that the installation and processing of these files by a user are the same as those expected by the authors. These two aspects are basic for any so4ware that wants to be published.

Also, he suggested some minor changes to the text in the uploaded pdf.

Ideally, it would be good if you can ask a colleague of yours to install it on a different computer and test if things run as expected elsewhere (I know... there is always something unexpected, as I also have worked on desktop software and mobile apps - so its best to get some external feedback by someone following the steps - or at least try on a second computer. I am aware of the turn around deadlines, though)

Reviewer 1 ·

Basic reporting

In the attached pdf

Experimental design

In the attached pdf

Validity of the findings

In the attached pdf

Additional comments

In the attached pdf

Annotated reviews are not available for download in order to protect the identity of reviewers who chose to remain anonymous.
Cite this review as

---

## Round 0.3 · accepted · Accept

From what I see all comments by the reviewer have been addressed. Thank you. You may still need to check some misspelling (e.g. where Figure 1 should be added it says "moudles", instead of modules )

I am happy that you found the review process helpful, as indeed it is often very complicated to get detailed feedback on your work - especially if there aren't so many experts around. I am also happy about the detailed comments by both reviewers who participated (it took the journal quite some time to find people accepting in the first round).